# The Natural History of Cervical Cancer and the Case for MicroRNAs: Is Human Papillomavirus Infection the Whole Story?

**DOI:** 10.3390/ijms252312991

**Published:** 2024-12-03

**Authors:** Giovanni Palomino-Vizcaino, Evelyn Gabriela Bañuelos-Villegas, Luis Marat Alvarez-Salas

**Affiliations:** 1Facultad de Ciencias de la Salud, Unidad Valle de las Palmas, Campus Tijuana, Universidad Autónoma de Baja California, Tijuana 21500, Mexico; gpalomino@uabc.edu.mx; 2Laboratorio de Terapia Génica, Departamento de Genética y Biología Molecular, Centro de Investigación y de Estudios Avanzados del I.P.N., México City 07360, Mexico; evelyn.banuelos@cinvestav.mx

**Keywords:** papillomavirus, cervical cancer, HPV, microRNA, miRNA, virology, carcinogenesis, infection

## Abstract

MicroRNAs (miRNAs) are small non-coding RNAs (ncRNAs) that negatively regulate gene expression. MiRNAs regulate fundamental biological processes and have significant roles in several pathologies, including cancer. Cervical cancer is the best-known example of a widespread human malignancy with a demonstrated viral etiology. Infection with high-risk human papillomavirus (hrHPV) has been shown to be a causative factor for cervical carcinogenesis. Despite the occurrence of prophylactic vaccines, highly sensitive HPV diagnostics, and innovative new therapies, cervical cancer remains a main cause of death in developing countries. The relationship between hrHPV infection and cervical cancer depends on the integration of viral DNA to the host genome, disrupting the viral regulator E2 and the continuous production of the viral E6 and E7 proteins, which are necessary to acquire and maintain a transformed phenotype but insufficient for malignant cervical carcinogenesis. Lately, miRNAs, the tumor microenvironment, and immune evasion have been found to be major players in cervical carcinogenesis after hrHPV infection. Many miRNAs have been widely reported as deregulated in cervical cancer. Here, the relevance of miRNA in HPV-mediated transformation is critically reviewed in the context of the natural history of hrHPV infection and cervical cancer.

## 1. Introduction

Cervical cancer is likely the best-known human malignancy associated with a viral etiology. Extensive epidemiological, clinical, and experimental data have long indicated that persistent and immunologically unrestrained infection with high-risk genital human papillomaviruses (hrHPVs) is causally associated with the appearance of cervical cancer [1,2]. Although a necessary factor, hrHPV infection is insufficient for malignant carcinogenesis. Other cofactors include some sexually transmittable infections (Human Immunodeficiency Virus and *Chlamydia trachomatis*), smoking, multiparity, and long-term use of oral contraceptives [3]. According to GLOBOCAN 2020, 604,127 new cervical cancer cases were reported annually, including 341,831 deaths worldwide [4]. Although prophylactic vaccines, highly sensitive genital HPV detection methods, and state-of-the-art therapeutics are currently available to render cervical cancer a nearly preventable disease [5], cervical cancer continues as a common tumor in women from developing countries, where the majority of new cases/deaths occur [4,6]. Therefore, there is a pressing need for novel and affordable prophylactic, diagnostic, and therapeutic approaches for cervical cancer available to regions with limited public health facilities and budgets.

## 2. Cervical Cancer

Infection with HPVs usually leads to a wide range of subclinical and clinical lesions, ranging from asymptomatic infections to benign warts or papillomas on the skin and genital mucosa. Although many genital HPVs may be deemed as part of the normal microbiota, long-term persistent infection with particular HPV types increases the risk for oncogenic progression [2,7]. Therefore, it is generally accepted that the onset of cervical carcinoma (CC) starts with HPV infection of the genital mucosae producing warts (condylomata) or atypical squamous cells of undetermined significance (ASCUS) and develops into progressive cytological alterations characterized by increased cell division and the accumulation of genetic changes in precursor dysplasia known as low-grade squamous intraepithelial lesions (LSILs) [8], which show a hyperproliferative state restricted to the lower part of the epidermis. In developed countries, treatment for LSILs is restricted to ablative or excisional procedures (i.e., laser ablation or electrosurgical excision) that destroy or remove cervical tissue [9]. At these early stages, cytological and histological screening plus HPV detection and typing may be crucial for a successful therapy.

Although most immunocompetent subjects will spontaneously clear both HPV infection and LSILs, if left unattended, some lesions will progress into high-grade squamous intraepithelial lesions (HSILs) characterized by the expansion of the proliferative cell layers to the upper part of the epithelium [8]. Severe HSILs may involve the full thickness of the epidermis and are sometimes referred to as in situ CC. About one-third of in situ CC patients may progress to CC, in which the neoplasia invades the stroma underneath, eventually reaching a metastatic state known as invasive CC. HSIL and CC rates remain higher than the threshold agreed by the WHO Cervical Cancer Elimination Initiative in countries associated with low levels of human development [10]. Therefore, the management of cervical cancer relies on the available resources for the determination of tumor stage, histology, and patient factors that may lead to an effective combination of therapies, including surgical approaches, radiotherapy, chemotherapy, and novel methods such as nucleic acid-based drugs, photodynamic therapies, therapeutic vaccination bevacizumab, immune checkpoint inhibitors (i.e., Anti-PD-1), and antibody–drug conjugates [11]. Currently, global initiatives aim to eliminate cervical cancer as a public health threat, focusing on advancement in imaging modalities, surgical approaches, histopathological risk factors, radiotherapy techniques, and biomarker-driven personalized therapies. Notably, the use of Anti-PD-1 has made a remarkable impact in the treatment of cervical cancer, leading to significant survival rates in advanced and invasive CC [12,13].

## 3. MicroRNAs (miRNAs)

### Biogenesis and Function

MicroRNAs (miRNAs) are highly conserved small (19 to 24 nt) non-coding RNAs (ncRNAs) that negatively regulate gene expression through hybridization with the 3′-UTR (untranslated region) of their target mRNAs [14]. The miRNA:mRNA interaction generally results in transcript degradation or translational arrest. All miRNAs are generated through a complex biogenesis process that involves several steps, starting from the transcription of miRNA loci within the cell genome to the processing and maturation of miRNA transcripts in the nucleus and cytoplasm. Most miRNA genes are transcribed by the RNApol II, producing long (>100 nt) hairpin-shaped primary miRNA transcripts (pri-miRNAs) containing a 5′-m^7^G cap and a 3′ poly(A) tail [15]. Although pri-miRNAs are regularly found in introns, they can either share the promoter with the host gene, or they can be independently transcribed from their own promoters [16], organized as single transcriptional units or in bicistronic or even polycistronic clusters [17]. 

The canonical pathway miRNA biogenesis begins in the nucleus after the transcription by RNApol II. The pri-miRNA is processed by the nuclear *Microprocessor* complex formed by DROSHA and DGCR8 proteins that removes the 5′ and 3′ ends to produce 60–70 nucleotide-long precursors with hairpin loop structures called premature miRNAs or pre-miRNAs [18]. DROSHA is a nuclear RNase III-like endonuclease that cleaves the pri-miRNA, while DGCR8 is a double-stranded RNA binding protein (RBP) that, through a heme group, functions as a molecular anchor recognizing the pri-miRNA by the loop, ensuring a correct cleavage by DROSHA [19]. Pre-miRNAs are then exported to the cytoplasm through the RanGTP-dependent nuclear transport reporter exportin 5 (XPO5) [20]. Cytoplasmic pre-miRNAs are cleaved by the RNase III-type nuclease DICER linked to the TRPB protein (transactivation-responsive RNA-binding protein) [21,22]. The crop by DICER is generated near the terminal loop, resulting in 21 nt-long mature miRNA/miRNA* duplexes [23], which are recognized by Argonaut (AGO) proteins and loaded into a ribonucleoprotein complex known as the RNA-induced silencing complex (RISC) [24]. Subsequently, the duplex is unwound into two single strands by RNA helicases [25]. The guide strand (miRNA mature strand) remains bound to RISC, whereas the passenger strand (miRNA*) undergoes degradation, forming the effector complex known as the miRNA-containing RNA-induced silencing complex (miRISC) [26]. Within this complex, the mature miRNA strand binds through its “seed sequence” to the 3′ UTR of the targeted mRNA. The “seed sequence” is crucial for the recognition of the mRNA target by the miRNA and comprises nucleotides 2 to 8 from the miRNA 5′ end [27].

The base complementarity between the miRNAs and their target transcripts determines the inhibitory pathway for translation. Full complementarity induces transcript degradation. However, partial complementary commonly results in translational repression or inhibition [28]. Therefore, a single miRNA can bind to multiple target mRNAs and regulate their functions [29], while any individual transcript may hybridize with several different miRNAs [30]. Approximately one-third of the protein-coding genes are regulated by miRNAs [31]. Overall, miRNAs regulate fundamental biological processes, such as proliferation, differentiation, inflammation, apoptosis, the cell cycle, and immune responses [32,33,34,35,36,37]. Due to their wide range of functions, miRNA dysregulation has significant roles in several pathologies, including cardiovascular diseases, neurological disorders, autoimmunity, and cancer [38]. Because miRNAs can target a large number of transcripts, the biological function of miRNAs may either produce the repression of an entire target network or silence only a few dominant targets [30].

## 4. Human Papillomaviruses (HPVs)

HPVs are small, non-enveloped, double-stranded DNA viruses that infect epithelial cells of the skin and mucosae and actively replicate in differentiating human keratinocytes. Genital HPVs belong to the genus *Alphapapillomavirus* of the *Papillomaviridae* virus family and are characterized by a circular DNA genome of approximately 8 kb in size [39]. A sub-group of genital HPVs types, including 16, 18, 31, 33, 35, 39, 45, 51, 52, 56, 58, and 59, has been causally associated with the appearance and persistence of preneoplastic and neoplastic lesions, including cervical, oral, anal, vulvar, penile, and head and neck tumors, hence termed “high risk” (hrHPV) [40]. HPV16 is the genital hrHPV type most commonly associated with latent infections, benign condylomata (warts), LSILs, HSILs, and half of invasive CC worldwide [41], and it is, therefore, the reference for hrHPV-associated cervical carcinogenesis. “Low-risk” HPV (lrHPV), such as HPV6 and HPV11, is associated with benign lesions, such as genital warts, laryngeal papillomas, and other mucosal lesions [42]. The comparison of the genomic and molecular features from HR- to lrHPVs has provided us with a deep understanding of the involvement of their encoded proteins in cervical carcinogenesis [43]. Consequently, the biological activities that are specific to hrHPV may account for their carcinogenic potential. Interestingly, the physical state of HPV DNA plays an important role in tumor progression. In condylomata and LSILs, the hrHPV genome is mainly found in episomal replicative forms, leading to productive infections [44]. However, in the majority of cervical tumors, hrHPV DNA is entirely or partially integrated into the host cell genome. This integration effectively interrupts the viral replicative cycle by disrupting genes regulating viral transcription and replication but keeps undergoing viral oncogenes expression. Consequently, only a limited number of hrHPV genes are constitutively expressed in CC [45].

### 4.1. HPV Genomic Structure

All genital HPVs locate their ORFs in one strand and are organized in three main genomic regions: the early (E) region containing genes controlling cell host and viral expression and propagation; the late region (L) harboring the genes coding for the viral capsid; and a long control region (LCR), sometimes referred as the non-coding region (NCR), containing the cis-regulatory elements required for viral transcription and replication. The viral genomic regions are bordered by early (A_E_) and late (A_L_) polyadenylation sites (Figure 1A). The E region spans over 50% of the HPV genome and encodes genes E1, E2, E4, E5, E6, and E7, which translate into individual proteins. Two other ORFs, E4 and E8, were initially assigned to this region, but only the E8 ORF from genital HPV31 encodes a spliced E8^E2C fusion protein, which functions as a negative regulator for viral transcription and replication [46], and the E4 protein derives from the spliced E1^E4 transcript and functions in the late phase of the hrHPV viral replicative cycle [47]. The L region covers about 40% of the viral genome and lies directly downstream of the E region, encoding the minor (L2) and major (L1) capsid proteins. The latter is well conserved among all papillomaviruses and constitutes the basis for HPV classification. An *L1* sequence identity of 60–70% defines a *Papillomaviridae* genus, which is named after a letter of the Greek alphabet (i.e., *Alphapapillomavirus*, *Betapapillomavirus*, *Gammapapillomavirus*, etc.). HPVs with 71–89% *L1* identity are considered a type and named in sequential numbering from their original description (i.e., HPV16, HPV18, HPV31, etc.). Over 200 different HPV types have been identified to date. HPVs with 90–98% *L1* identity are defined as subtypes, and those with >98% *L1* sequence identity are designated as variants [48,49,50,51].

### 4.2. The Long Control Region (LCR) and HPV Transcription

The LCR is a non-coding segment of approximately 800 bp (10% of the HPV genome) situated between the L2 and E6 loci. Despite the high sequence variability among all genital HPV LCRs, this region harbors common regulatory and tissue-specific elements distributed in three domains relative to the start codon of the E6 gene: the distal domain, the keratinocyte-specific (KE) domain, and the proximal domain [52]. In HPV16 and 18, the distal domain contains a binding site for the E2 viral protein (E2BS) plus the transcription termination and A_L_ sites for L1 and L2 transcripts. The KE domain harbors multiple transcription factor binding sites for the RNApol II-directed transcription of the viral early and late promoters, including AP-1, NF-1, TEF-1, OCT-1, YY-1, BRN-3a, NF-IL6, KRF-1, NF-kB, FOXA1, and GATA3 transcription factors, among many others [52,53]. The proximal domain contains three E2BS for initiating replication and transcription control and several binding sites for basal transcription factors, such as Sp-1, the TATA box, and the early promoter [54]. The proximal domain also contains an origin of replication (*ori*), which is required for HPV replication [55].

**Figure 1 ijms-25-12991-f001:**
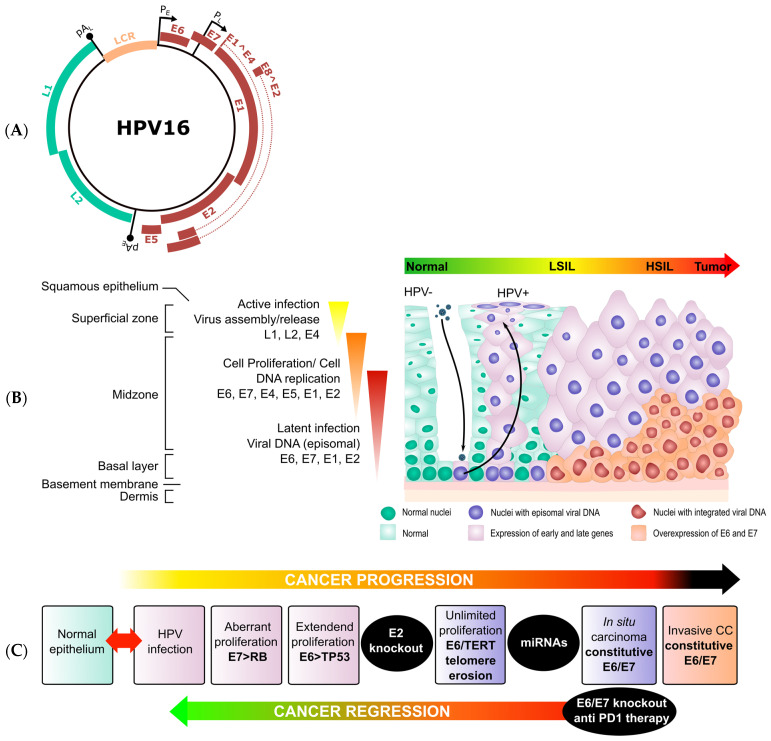
HPV16 genome. (**A**) The HPV16 early genes E1, E2, E4, E5, E6, E7, and E8 are marked in red, and the capsid genes L1 and L2 are marked in green. Relevant viral transcriptional elements, such as the early (P_E_) and late (P_L_) promoters, early (pA_E_) and late (pA_L_) polyadenylation sites, and the spliced transcripts E1^E4 and E2^E8, are indicated [56]. (**B**) Natural history of hrHPV infection and cervical carcinogenesis. The progression from hrHPV infection towards a CC through LSIL and HSIL is schematized in the tisular context of the cervical epithelium. (**C**) Key molecular events in cervical carcinogenesis from hrHPV infection to an invasive CC.

The hrHPV early promoters are generally located immediately downstream of the TATA box and named after the genomic sequence number corresponding to the 5′ end of the E6 gene (hence P_97_ for HPV16 and HPV31 or P_105_ for HPV18) [53]. Because there is no antisense transcription, hrHPVs transcribe their genes as polycistronic transcripts with a single 5′-m^7^G cap devoid of 5′ UTR internal ribosome entry sites (IRESs) and containing alternative splicing donor and acceptor sites that produce several transcripts with ORFs in the three possible reading frames [57]. In genital lrHPVs (i.e., HPV6), there are two alternative early promoters to express E6 from one promoter and E7 from a further downstream promoter [58].

The late promoters of hrHPVs are usually located in or around the *E7* gene sequence (i.e., P_670_ in HPV16) and are regulated by a number of transcription factors specific to late keratinocyte differentiation, including AP-1, CDP, C/EBP, and SKN-1a [59]. Late transcription is enhanced by the viral vegetative DNA replication in terminally differentiated keratinocytes. Late viral transcripts are also alternatively spliced to produce isoforms of L1 and L2 mRNAs [56,60,61].

Translation of early hrHPV mRNAs occurs from the first AUG. Consequently, translation is most efficient for the ORFs closer to the 5′ end of the early transcripts (i.e., E6, E7, and E1), while weakly translatable downstream ORFs may turn into the 3′ UTR. Because hrHPV early mRNAs exhibit very short, if any, leader sequences, they utilize alternative RNA splicing to individually express E6 and E7 from the single early promoter. Alternatively, pA_E_ and pA_L_ sites are used in combination with alternative RNA splicing, thus allowing L1 and L2 gene expression from the late promoter. Moreover, the various translational frames generated by alternative RNA splicing permit E1^E4 or E8^E2 expression from transcripts with the ORFs partially overlapping E2 or E1, respectively. Consequently, the hrHPV transcriptome consists of a relatively large array of alternatively processed RNAs, allowing an appropriate level of individual viral proteins at specific keratinocyte differentiation stages to attain productive virus infection. The disruption of the highly complex hrHPV transcription has been closely associated with cervical carcinogenesis [39].

### 4.3. The Replication of hrHPV: The E1 Protein

HPVs have been co-evolving with their primate hosts even before the emergence of *Homo sapiens*, although their mutation rate is extremely slow [62,63,64]. Because HPVs lack their own DNA polymerases, the evolutionary pathway of hrHPVs relies on the high-fidelity host DNA polymerases and the multiple replicative cycles of the virus during a productive infection [65]. The replication of the hrHPV genome is a tightly regulated process controlled by the viral E1 and E2 proteins. E1 is the largest (approx. 600–650 amino acids) and most conserved protein encoded by hrHPVs. It functions as an ATP-dependent hexameric DNA helicase and is the only enzyme coded by papillomaviruses. The E1’s high degree of conservation may be an indication of the crucial role it plays as the replicative helicase for the initial and productive replication of the viral genome within the nucleus of infected keratinocytes [66]. The domain structure of E1 includes a bipartite nuclear localization signal (NLS), a DNA-binding domain (DBD), and the helicase domain (HD), which includes a minimal oligomerization domain (O), the ATP-binding domain, and a C-terminal brace [67].

For viral DNA synthesis, E1 must initially recognize the specific *ori* (mapped to the 3′ portion of the LCR) consisting of two E2BS, a palindromic E1-binding region, and an AT-rich sequence [68]. E1 DBD recognizes low-affinity specific sequences at the Ori consisting of up to six E1-binding sites (E1BS 1–6) of the consensus sequence: 5′-AT(A/G/T)G(C/T)(C/T)-3′ [69]. Nevertheless, the critical step in the initiation of viral DNA replication is the ATP-dependent assembly of E1 as a double-hexameric helicase capable of unwinding the *ori* and the DNA ahead of the replication fork. E1 also engages in numerous interactions with specific host factors to direct the assembly of a functional replisome required for the bidirectional replication of the viral genome in the epithelial basal layer [70]. Stable recognition of the *ori* by E1 requires the viral E2 protein to initiate DNA replication [71,72]. The role of E2 during replication is to recruit E1 specifically at the *ori* by bridging the binding between E1 and the E2BS located at the *ori*. The E1–E2–*ori* complex then becomes a template for the assembly of the E1 double-hexamer with unwinding activity [73]. Additionally, E1 recruits the host cell DNA polymerase α-primase, the topoisomerase I complex, and the replication protein A to complete a replication fork [70,74,75].

Although E1 and E2 support transient replication of episomes containing the minimal Ori, the persistent and stable maintenance of hrHPV genomes requires more than one E2BS in the LCR [76]. For HPV-16, 18, and 31, three E2BS sites are sufficient for the maintenance of their genomic replicons [50,77]. The E2 protein links the multiple E2BS on the viral genomes to the host chromosomes in mitosis through protein–protein interactions, thus enabling the maintenance and partitioning of the viral episomes [78,79]. The best-known example of an E2 tethering target is the chromatin adapter protein BRD4 [80]. Dissociation of the E2-BRD4 complex results in the displacement of viral episomes from chromosomes [81]. Although all hrHPV E2 proteins bind BRD4, they do not appear tightly bound to mitotic chromosomes [82].

### 4.4. HPV Transcription Control: E2 Protein

The E2 protein can be considered the master regulatory protein for the hrHPVs replicative cycle. Notwithstanding the essential functions of the E2 protein in the replication and maintenance of the viral genome, the main function of E2 is as a transcriptional regulator. Although lacking the structural complexity of E1 and the numerous and complex cellular interactions of E6 and E7, the E2 protein performs major regulatory roles in both viral transcription and replication. In all papillomaviruses, the E2 proteins are specific DNA-binding proteins that interact with the various 12 bp E2BS (5’-ACCGN_4_CGGT-3′) located at the LCR. The hrHPV full-length E2 protein is about 300 amino acids long and consists of a conserved N-terminal transactivation domain (TAD) linked to a C-terminal DNA-binding/dimerization domain (DBDD) through a flexible hinge linker sequence [83,84]. As stated above, E2 assists E1 to initiate and maintain viral DNA replication but also promotes the retention of viral genomes in the nucleus and their tethering to host chromosomes to guarantee viral DNA distribution into the daughter cells [85]. Through specific binding to the E2BS present in the LCR, E2 proteins activate or repress transcription depending on the number and relative location of these binding sites and the nature of the interacting transcription factors. In hrHPVs, two E2BS are positioned near the viral TATA box for the early promoter. Upon E2 binding, basal transcription factors, such as Sp1 and TBP, are sterically displaced, thus repressing early transcription [86,87]. E2 also represses transcription by recruiting chromatin remodeling factors [88].

In hrHPVs, the E2 gene can also be transcribed from both the early and late promoters. In the lower layers of an infected epithelium, E2 mRNAs are transcribed from the early promoter and terminated at pA_E_. Once the infected keratinocyte starts differentiation, the late promoter is activated but still produces full-length E2 transcripts truncated at pA_E_ via differential splicing [60]. In HPV-31, an E8^∧^E2 transcript from the early promoter produces a truncated E2 protein without the TAD that functions as a transcriptional repressor [89]. This variety of E2 transcripts allows for a relative abundance of E2 proteins in differentiated keratinocytes, thus providing a molecular environment suitable for high-level viral vegetative replication.

E2 exhibits a strong anti-proliferative action when ectopically expressed in CC cells, inducing both G1 cell cycle arrest and apoptosis [90]. While the cell cycle arrest is explained by the E2-mediated transcriptional repression of the viral oncogenes E6 and E7, the induction of apoptosis is an independent function of E2 mapped to the amino-terminal transactivation domain (TAD) of the E2 protein not linked to transcriptional transactivation. E2 induces apoptosis through the extrinsic pathway, involving the initiator caspase 8 [91].

### 4.5. Mastering the Host Cell: The hrHPV Oncogenes E5, E6, and E7

E5 is a small, highly hydrophobic transmembrane protein that, in dimeric form, interacts and activates tyrosine kinase receptors, including the EGF-R and PDGF-R [92]. The best-known E5 protein is encoded by the bovine papillomavirus type 1 (BPV1). BPV1 E5 has long been considered an oncogene because of its natural ability to independently transform cultured murine fibroblast cell lines (NIH3T3 and C127) and non-immortalized human foreskin fibroblasts [93,94]. In bovine cutaneous warts associated with BPV1 infection, E5 protein is localized to the basal keratinocytes and to terminally differentiated keratinocytes in the upper epithelial layers where virion production occurs [95]. Similar activities have also been shown for hrHPV E5 proteins, which are multi-pass transmembrane proteins with limited sequence identity with BPV1 E5. However, the hrHPV E5 is twice the size of BPV1 E5, with weak transforming activity in cultured cells and animal models [96,97]. Despite extensive analyses, the molecular basis for hrHPV E5 oncogenic activities in genital mucosae is poorly understood as the hrHPV integration into the host genome in CC cells disrupts the continuity of the E1 and E2 genomic loci [98], with the concomitant loss of all of the *E5* gene [99]. Moreover, the hrHPV E5 function appears otherwise related to immune evasion during a productive infection by downregulating the cell surface proteins involved in antigen presentation. HrHPV E5 binds with the heavy chain component of the human MHC-I antigen (HLA-I), causing retention of HLA-I in the Golgi apparatus and ER, thus blocking its transport to the cell surface [100,101].

Clinical and molecular data have shown that the hrHPV E6 and E7 proteins are key factors associated with the appearance and progression of cervical tumor cells. E6/E7 are found and expressed in most CC tumor cells [102]. E6 and E7 induce aberrant, extended, and unlimited proliferation (immortalization) in several cell types, including cervical keratinocytes [103,104,105,106]. Specific silencing of hrHPV E6/E7 expression results in apoptosis induction and the inhibition of cell proliferation and senescence [107,108,109,110,111]. Furthermore, the expression of hrHPV E6/E7 has been shown to be necessary for tumor formation of hrHPV-containing cells in xenograft and transgenic mice models [112].

Many clinical and experimental setups have shown the diverse phenotypic effects induced by the hrHPV E6/E7 oncoproteins, including those defined as “hallmarks of cancer”, such as sustained proliferative signaling, evasion of growth suppressor control, resistance to apoptosis, replicative immortality, angiogenesis induction, activation of invasion and metastasis, genome instability, inflammation, reprogramming of energy metabolism immune evasion destruction, and the generation of a surrounding microenvironment favorable to tumor growth [113].

HrHPV E6 and E7 proteins are relatively small proteins with a length of approximately 150 and 100 amino acids, respectively. Although no intrinsic enzymatic activities have been reported, these proteins manipulate the keratinocyte cell cycle through the formation of complexes with cellular proteins. Systematic interactome analyses led to the identification of a high number of novel cellular proteins that could interact with E6 or E7, although the biological significance of many of these interactions is yet to be described [114,115]. Nevertheless, the primary activities of E6 and E7 proteins relate to the binding and inactivation of the TP53 and Retinoblastoma (RB) tumor suppressors, respectively.

The hrHPV E6 protein was first shown to interact with the TP53 tumor suppressor protein [116] and the E6-AP complex to induce the specific ubiquitination and degradation of TP53 [117,118], thus hindering TP53-mediated apoptosis [119]. E6 has also been reported to interact with the extrinsic apoptotic factors TNFR-1, FADD, and caspase-8, suggesting alternative apoptosis inhibitory functions [120,121]. Although E6-mediated degradation of TP53 may be considered a key event for the onset of cellular transformation, E6 possesses other TP53-independent transforming and anti-apoptotic activities, such as telomerase activation [122,123,124]. Other cellular targets of hrHPV E6 proteins include PDZ domain-containing proteins, such as the human homolog of the tumor suppressor DLG (discs large protein) [125], MUPP1 [126] and MAGUK (membrane-associated guanylate kinase) [127] proteins, and a number of transcription regulators [128], disrupting cell adhesion, polarity, and epithelial differentiation and reducing immune recognition of HPV-infected cells [129]. HrHPV E6 also binds to the cellular ubiquitin ligase E6AP (E6-associated protein), leading to a conformational shift of E6, which allows the formation of a trimeric E6/E6AP/TP53 complex [119], resulting in the proteolytic degradation of TP53 [130].

The E7 protein plays a vital role in the viral replicative cycle by disrupting the tight link between differentiation and proliferation, thus allowing viral replication in normal keratinocytes that would be otherwise withdrawn from the cell cycle [131]. E7 protein from hrHPVs targets RB and disrupts the E2F-mediated transcriptional regulation, resulting in the upregulation of genes required for G1/S transition and DNA synthesis [132]. HPV-16 E7 can directly bind the G1/S transition antagonists E2F1 [133] and E2F6 [134], thus ensuring that the infected cells remain in an S-phase-competent state and allowing HPVs to bypass negative growth signals. The steady-state level and metabolic half-life of RB are decreased in HPV-16 E7-expressing cells because E7 can induce the degradation of RB through the ubiquitin–proteasome system [135,136]. HrHPV E7 proteins also contribute to cell cycle dysregulation through the abrogation of the growth inhibitory activities of p21^CIP1^ and p26^KIP1^ [137,138,139]. Other functions associated with hrHPV E7 expression include epigenetic reprogramming through induction of KDM6A and KDM6B histone demethylases [140], trophic sentinel signaling abrogation and autophagy induction [141], induction of genomic instability [142], and disruption of Anoikis signaling through interaction with p600 [143].

The transforming activity of E7 in the foreskin and cervical keratinocytes is significantly stronger when E6 is co-expressed [144,145,146]. The simultaneous targeting of TP53 and RB by the viral oncoproteins could provide an explanation for this functional cooperativity. Specifically, abnormal proliferative signals, such as those exerted by activated oncogenes, can result in TP53-mediated induction of apoptosis or senescence, thereby protecting multicellular organisms against the emergence of growth-deregulated cells [147]. Oncogenic HPVs could allow the cells to escape from this anti-tumorigenic defense mechanism, since the abnormal growth stimulus resulting from E7-mediated RB inactivation cannot be counteracted by TP53, which is degraded via E6. In line with this model, E7, when expressed alone, induces TP53 and has pro-apoptotic potential [138], whereas the specific inhibition of E6 (in the presence of ongoing E7 expression) is linked to the induction of apoptosis in cervical cancer cells [148,149].

### 4.6. The Way of Infection: hrHPV Capsid Proteins

The papillomavirus non-enveloped virion is an icosahedral capsid of (T = 7) 55–60 nm diameter consisting of 360 copies of L1 assembled in 72 capsomers and about 12–60 molecules of the L2 protein per virion buried underneath the capsid surface and exposing only the amino-terminal residues to the surface. Sixty of the L1 capsomer subunits are at hexavalent positions, interacting with six neighboring capsomers, with the remaining 12 capsomers at pentavalent positions [150,151]. Capsomers are interconnected by the disordered and flexible C-terminal of L1 molecules. Additional stability is provided by L1 forming intercapsomeric disulfide bonds amongst conserved cysteine residues [152,153]. The L1 protein can interact with heparan sulfate proteoglycans (HSPGs) located at the extracellular matrix and on the target cell surface [154]. A remarkable feature of the L1 protein is that it can self-assemble into virus-like particles (VLPs), even with structural and immunological properties identical to the natural virions [155,156].

The precise amount, location, and orientation of L2 within the mature hrHPV capsid are still to be defined [157,158]. A number of cellular proteins have been identified as L2 protein ligands, including Cyclophilin B, Syntaxin 18, and sorting Nexins 17 and 27 [159,160]. Moreover, L2 also harbors a transmembrane domain (TMD) plus nuclear export (NES) and nuclear localization (NLS) sequences that are essential for the capsid intracellular trafficking [161] and a chromosome-binding region (CBR) containing a highly conserved SUMO-interacting motif (SIM) [162,163]. When co-expressed in the presence of L1 and suitable covalently closed circular DNA (cccDNA), L1 and L2 spontaneously form pseudoviruses (PsVs) and quasiviruses (QsVs), which can mimic the hrHPV infection [164,165].

HrHPV infection into actively mitotic keratinocytes in the epithelial basal layer requires the interaction of the viral L1/L2 capsids with heparan sulfate proteoglycans (HSPGs) and laminin-332 molecules found in the extracellular matrix (ECM) or to heparan sulfate (HS) residues directly on the cell surface [166,167]. Such interaction results in conformational changes in both proteins and the subsequent transfer of the virion to a cell surface entry receptor complex [168]. HSPGs are essential for hrHPV infection, and at least three HS binding sites have been identified on the HPV16 capsid [169]. While on the entry receptor complex, kallekrein-8 (KLK8) cleaves the L1 protein, allowing the emergence of L2 on the virion surface, and the additional action of the cyclophilin B (CyPB) exposes the N-terminus of L2 [159]. The processed capsid with exposed L2 becomes susceptible to cleavage by the Furin convertase, permitting, in turn, the adequate subcellular trafficking of L2 with the viral DNA [170]. Virion entry occurs through the induction of actin dynamics signaling involving tetraspanin CD151 associated with α3β1, α6β1, and α6β4 integrins, growth factor receptor tyrosine kinases (GFTKs), ANNEXIN A2, and the cytoskeletal adaptor obscurin-like 1 (OBSL1) [171]. This process induces the formation of non-coated vesicles that associate with trafficking adaptors that facilitate early endosome (EE) maturation and virus trafficking to late endosome/multivesicular bodies (LE/MVBs). There, acid-dependent cathepsin within the endolysosomal compartments triggers the uncoating of the virions towards the continuation of intracellular trafficking into the nucleus [172].

The C-terminus of L2 harbors a cell-penetrating peptide (CPP) that allows the incorporation of the L2 TMD into the vesicular lipid bilayer, while the luminal N-terminus and most of the protein remain exposed to the cytosol [173,174]. The viral genome and remnants of the capsid remain inside the non-coated vesicle, thus evading innate immune detection [175]. The cytosol-exposed part of L2 connects the vesicle to the cellular transport machinery through interactions with components of the retromer complex to complete endosome tabulation and the retro-transportation from the endosome to the trans-Golgi network (TGN) [176]. The hrHPV DNA and the remnants of the capsid remain at the TGN until the onset of mitosis. The fragmentation of the TGN during mitosis promotes the release of virion-loaded vesicles, thus enabling the subsequent transport into the nucleus [177]. After dissociation from the TGN, the virion–vesicles bind to microtubules associated with the dynein motor complex through a consensus Tctex interaction domain located in the C-terminus of L2 [178,179]. In addition, L2 also forms a complex with the Ran-binding protein 10 (RanBP10) and karyopherin alpha 2 (KPNA2) to promote the minus-end-directed transport towards the mitotic chromatin [180]. While the viral DNA and the remnants of L1 remain inside the vesicles, L2 tethers to mitotic chromosomes through the CBR, enabling the inclusion of the virus into the newly formed nucleus [162,181]. After mitosis, the L2 SIM directs the loading of the viral genome into newly formed promyelocytic leukemia nuclear bodies that enable viral transcription and replication while impeding the viral LCR from interaction with repressive transcription factors such as Sp100, MYPOP, Tbx2, and Tbx3 [182,183,184,185,186].

### 4.7. The HPV Replicative Cycle

Productive hrHPV infection heavily relies on replication and transcription factors present in keratinocytes from stratified squamous epithelia (Figure 1B). The lack of viral DNA and RNA polymerases and the multiple functions and interactions displayed by hrHPV proteins are a direct reflection of the long and extremely successful co-evolution with *H*. *sapiens* [187]. All papillomaviruses (including hrHPVs) manage the host cell resources to produce a huge number of replicas while evading the host’s adaptive and innate immune responses. Nevertheless, hrHPVs require proliferative keratinocytes for successful infection and replication and, therefore, must gain access to the epithelial basal layer across several physical barriers. The mucous formed by the glandular secretion of viscous fluid containing enzymes and extracellular antimicrobial factors, along with several layers of non-dividing cells, can impede access to mitotically active keratinocytes, which are required for successful infections. The resilience of hrHPV capsids provides for opportunistic infection through the squamocolumnar junction in the cervix and epithelial microtrauma or permeabilization as they enable viral access by exposing L1 HSPGs on the basal membrane or the cell surface of basal layer cells to capsid L1 and initiate infection [161,188,189]. HrHPV virions penetrate the cell membrane and are endocyted through L2-mediated interactions with the TGN pathway and the dynein motor complex to the nucleus in mitotic keratinocytes as described above.

Once in the nucleus, hrHPVs transcribe only early genes E6, E7, E1, E2, and E5 (in genomic order) from the early promoter using cellular tissue-specific transcription factors and the RNA polymerase II transcription machinery. The hrHPV DNA replication process occurs in two stages with the obvious aim of producing more viruses while maximizing the probability of a productive infection from minimal infection events. An initial DNA replication step relies on the function of E1 and E2 proteins hijacking DNA polymerases and cellular replication factors to produce episomal copies of the hrHPV genome at about 50–100 copies per infected cell by bidirectional replication [190,191]. Such “maintenance” replication occurs, on average, once per cell cycle during the S phase and may persist for years [39]. Interestingly, E2 also interacts with BRD-4 and may bind to the host chromosomes, thus assuring the distribution of the newly replicated viral genomes into the daughter cells [192]. The E5, E6, and E7 proteins stimulate host cell division, providing an environment rich in host cell DNA replication factors and polymerases and inhibiting apoptosis and senescence while assisting immune evasion. Noteworthy, this stage is space-time-limited as the E2 protein accumulates during the proliferative basal keratinocyte status and eventually shuts down the early promoter inhibiting E6 and E7 expression, thus ceasing cell hyperproliferation and allowing for keratinocyte terminal differentiation. The late promoter mediates several early viral genes expression, including E1, E2, and E4. However, E2 induces the production of L1 and L2 from the late promoter and allows for viral DNA replication at a vegetative rate (>1000 copies per cell) through a rolling circle replication function [190]. When and how the switch from maintenance to vegetative hrHPV replication occurs and its relationship to the keratinocyte terminal differentiation program remain open questions.

Following the keratinocyte terminal differentiation program, L1 and L2 proteins self-assemble encapsidating individual hrHPV genomes, forming infectious virions, which accumulate within the keratinocyte until squames (heavily keratinized dead cells lacking a nucleus or other organelles) are formed in the stratum corneum and eventually released [44]. During this process, the E4 protein is expressed at high levels before L1 and L2 contribute to genome amplification and virion synthesis through their association with and remodeling of the cellular keratin network, suggesting plausible roles in virus release and/or transmission [47].

## 5. HPV and Cervical Cancer

As stated above, experimental and clinical evidence indicate that the hrHPV E6 and E7 proteins are essential to maintaining the malignant phenotype of cervical cancer cells [193]. Although the E6 and E7 oncoproteins’ capacities to, respectively, bind and inactivate the TP53 and RB tumor suppressor pathways are fairly understood within the context of the hrHPV replicative cycle, their role in cervical malignant transformation is still unclear. Furthermore, the most significant molecular event for the hrHPVs’ relationship to cervical cancer appears to be the disruption of E2 expression. In the absence of the E2 protein, hrHPV E6 and E7 genes are continuously transcribed from the early promoter under the control of cellular transcription factors interacting with the viral LCR, leading to the constitutive production of E6 and E7 proteins [194]. The hrHPV E2 protein can efficiently block E6/E7 transcription through the two E2BS located close to the early promoter [89,195]. Noteworthy, most cervical tumor genomes contain integrated copies of hrHPV DNA interrupting the viral E2 gene [196]. Such integrations also disrupt E1, E4, E5, L1, and L2 genes, rendering E6 and E7 as the only viral fully expressed genes in cervical tumors [197]. Additionally, in tumors containing episomal hrHPV DNA (with intact E2 ORF), access of the E2 protein to the LCR may be inhibited by methylation of the E2BS [198], viral integration resulting in tandem copies of hrHPV genomes that increase E6/E7 expression [199], or even the production/stabilization of E6/E7 transcripts from adjacent host DNA sequences [199].

Thus, the established hypothesis of hrHPV-mediated cervical carcinogenesis proposes that after successful infection and E2 disruption events, the hrHPV E7 protein initiates the carcinogenesis process by inactivating RB, and the subsequent release of E2F transcription factors causes TP53 activation [200,201] (Figure 1C). This is a well-known cellular response that normally results in G1/S cell cycle arrest and/or apoptosis [202,203]. Concomitantly, E6 binds the UBE3A ubiquitin ligase complex and tags TP53 for ubiquitination and proteasomal degradation, and, therefore, both hrHPV E6 and E7 are required for aberrant and extended cell proliferation [204]. The anomalous proliferation of hrHPV E6/E7-containing cells produces telomere erosion, which is thwarted by E6-mediated activation of telomerase, thus leading to unlimited proliferation (immortalization) [205,206]. Such a scenario renders E6/E7-containing cells defenseless against external mutagens, high genomic instability, chromosomal defects, Anoikis resistance, etc., as the causes of cervical malignant transformation.

Nevertheless, this hypothesis does not account for the additional TP53- and RB-dependent and -independent tumor pathways nor the response of the interferon-stimulated host apolipoprotein B mRNA-editing enzyme catalytic polypeptide-like 3 (APOBEC3) family of cytidine deaminases. It has been shown that APOBEC3 fights infection by introducing deleterious mutations into the viral DNA genome, causing viral clearance either through specific changes to activate an immune response or by accumulative lethal mutagenesis [207]. Moreover, the accumulation of E6 and E7 proteins resulting from E2 disruption reveals additional oncogene functions otherwise irrelevant for a productive hrHPV infection. E6 contains a PDZ binding motif (PBM) that binds to and induces proteolytic degradation of several cellular proteins with PDZ domains, including the potential tumor suppressor proteins MAGI-1, Dlg, and Scribble, and also impairs the PDZ protein function by altering their subcellular localization [208]. The hrHPV E6 protein also upregulates telomerase activity, averting telomere shortening and replicative senescence [209]. HrHPV E7 induces p16^INK4A^ independent of RB inactivation through the epigenetic activity of KDM6B over histone H3 (H3K27me3), which is fundamental for silencing by the Polycomb repressive complexes (PRCs) [210,211]. HPV E7 proteins also induce p14^ARF^, which is also silenced by PRCs. HPV16 E7 expression causes a decreased incidence of H3K27me3 marks on the p14^ARF^ promoter independent of p16^INK4A^ activation and KDM6B activity [210]. E6/E7 expression also affects the aerobic glycolysis of CC cells through IGF2BP2-mediated methylation of MYC mRNA [212].

The role of APOBEC3 variants (APOBEC3A and APOBEC3B) in HPV infection is not yet clear, but several deep sequencing reports on cervical tumors and exfoliated cervical cells appear to confirm APOBEC3-directed mutagenesis of hrHPV types [207,213,214]. Thus, APOBEC3 may help to reduce viral persistence and, consequently, cervical carcinogenesis progression. Nevertheless, APOBEC3’s role in controlling viral infection may also result in off-target mutagenesis over somatic host genes that may contribute to carcinogenesis [215]. In particular, in the highly replicative environment produced by hrHPV infection in the epithelial basal layer, it is likely that the effect of APOBEC3 on the single-stranded replicative forks or transcriptionally active viral genes may be extended to the host cell counterparts, thus leading to extensive mutagenesis.

On the clinical side, the vast majority of hrHPV-associated cervical LSILs and HSILs regress spontaneously, a phenomenon largely attributed to immune intervention [216]. However, in the USA alone, the estimated lifetime risk for hrHPV infection (up to 75%) [217] sharply contrasts with the lifetime risk for developing cervical cancer (∼0.7%), and cervical tumors occur many years after hrHPV (https://seer.cancer.gov/statfacts/html/cervix.html, accessed on 2 December 2024). Taken together, molecular and clinical data suggest that despite the easy transmissibility and the oncogenic functions of E6 and E7 proteins to produce a hyperproliferative phenotype, the hrHPV infection is a necessary step but not sufficient to produce cervical tumors. In addition, cervical cancer shows very high intratumor heterogeneity (Cancer Genome Atlas Research Network), but it relies on E6/E7 expression to maintain a hyperproliferative phenotype. In fact, as with E2 ectopic expression [218], specific targeting of E6/E7 mRNA through antisense oligonucleotides, catalytic nucleic acids, and siRNAs results in apoptosis induction and senescence in cervical tumor lines [45]. This reliance on E6/E7 expression for proliferation, regardless of insufficiency for a fully transformed phenotype, is often referred to as “addiction” [39].

HrHPV E6/E7-immortalized keratinocytes rarely progress into malignant transformation in the absence of external stimuli [219,220,221]. In fact, there is a lack of in vitro and animal models that fully recapitulate the natural history of cervical cancer. The few reported cell models for hrHPV carcinogenesis establish that once the continuous E6/E7 expression produces “addictive” hyperproliferation and apoptosis/senescence evasion, subsequent molecular events are required for a fully transformed phenotype. In agreement with this hypothesis, it has been shown that ectopic episomal HPV16 DNA in combination with ectopic expression of *MYC* and *PIK3CA^E545K^* genes conferred tumor-initiating ability of E6/E7-dependent enhanced clonogenic growth on late-passage telomerase-immortalized keratinocytes. Interestingly, additional ectopic expression of active *MEK1 (MEK1DD)* and/or *KRAS^G12V^* generated highly potent tumor-initiating cells [222]. In the last two decades, many miRNAs have been shown as key players in several cellular processes, such as apoptosis, proliferation, senescence, EMT, and differentiation (Figure 1C). However, their role in cervical carcinogenesis is not fully understood.

Interestingly, hrHPV E6/E7 can induce a senescent phenotype by impairing mTOR through hypoxia, highlighting the role of aging in cancer development, as seen in the age of incidence of CC. Aging is marked by a gradual decline in function, a characteristic of nearly all aging organisms. In multicellular organisms with renewable tissues, aging also involves gain-of-function changes that lead to hyperplasia. These changes, through genomic instability, enable cells to develop traits that enhance their proliferation, migration, colonization of new sites, survival in hostile environments, and evasion of the immune system. Besides halted growth, senescent cells accumulate with age, creating a microenvironment conducive to cancer initiation and progression. This is due to the senescence-associated secretory phenotype, where cells secrete various proinflammatory cytokines, chemokines, growth factors, and proteases to neighboring cells [223].

## 6. MiRNAs in Cervical Cancer

Because miRNAs are involved in most cancer types, and as tumor initiation and promotion in cervical cancer heavily relies on the unregulated functions of the hrHPV E6 and E7 proteins, it can be assumed that E6/E7 expression may cause disruption of the cervical miRNome. This can be attained by either adding viral miRNAs to the infected cell miRNome or by controlling the expression and function of cellular miRNAs [224]. Aside from RNA viruses, some DNA viruses encode their own miRNAs required for the regulation of the transition from latent to lytic gene expression or the control of viral transcription or replication regulators. However, hrHPV-coded miRNAs have yet to be identified, and viral transcription and replication control functions are largely accomplished by E1 and E2 proteins.

In other instances, DNA viruses may control the levels of cellular miRNAs. Such is the case for miR-27 silencing by herpesvirus saimiri (HVS) [225] or the degradation of the miR-17~92 cluster by human cytomegalovirus (CMV) [224]. On the contrary, miR-155 is upregulated by Epstein–Barr virus (EBV) [226]. Moreover, host miRNAs can control the lytic-to-latent switch in herpesvirus-infected cells, as is the case for miR-138 and miR-200, which promotes the latency of herpes simplex virus type 1 (HSV-1) and HCV, respectively. So far, the hrHPV replicative cycle has not been associated with any particular miRNA or miRNA pathway. However, bioinformatic and experimental data show that E6/E7 expression disrupts the cellular miRNome [227,228,229].

HrHPV E6 and E7 directly and indirectly interact with cellular transcription factors and epigenetic regulators that, in turn, cause alterations in gene expression. In addition to protein-coding genes, hrHPV E6 and E7 can also cause alterations in the expression of ncRNAs, including miRNAs [230]. A large number of miRNAs have been associated with the initiation, progression, and malignant transformation of CC. However, the relatively low number of CC without hrHPV infection suggests that the participation of miRNAs in cervical carcinogenesis largely resides in post-integration molecular events. Thus, the miRNome disruption in CC is a consequence of the sustained hrHPV E6/E7 expression. As TP53 and RB are the main cellular targets for E6 and E7, respectively, it is likely that genes responsive to TP53 and E2F would be affected after infection and remain silent after integration.

It has been shown that miRNA responders to TP53 have been consistently found to be downregulated through progression from normal keratinocytes to CC (Table 1) activation [231]. Of particular interest are the miR-34 (miR-34a/b/c), miR-192 (miR-192, miR-194, and miR-215), and miR-200 (miR-200a/b/c, miR-141, and miR-429) families, which are transcriptionally activated by TP53 [232,233,234]. All of these miRNAs are considered tumor suppressors through different targets in diverse pathways. The individual action of each one usually results in the inhibition of proliferation and apoptosis induction. As stated above, the inactivation of RB by E7 releases E2F transcription factors that would activate promoters with E2F binding sites, including miRNA promoters of miR-449 and the miR-17~92 cluster (miR-17, miR-18a, miR-19a/b, miR-20a, and miR-92a-1) [235] and its paralog clusters miR-106a~363 (miR-18b, miR-19b-2, miR-20b, miR-92a-2, miR-106a, and miR-363) and miR-106b~25 (miR-25, miR-93 and miR-106b) [236], which are referred to as pro-tumor miRNAs or oncomiRs [237]. Therefore, it would be expected that the sustained expression of hrHPV E6/E7 results in the downregulation of specific tumor suppressor miRNAs and the upregulation of a defined set of oncomiRs commonly reported as progressively upregulated during cervical carcinogenesis (Table 1).

However, an increasing number of transcriptome studies from CC-derived cell lines, condylomata, LSILs, HSILs, and CC tumors have detected expression anomalies in TP53- and E2F-responsive miRNAs. Overall, the compilation of several studies on the miRNome from HSIL and CC samples and cell lines showed that miRNA disruption in cervical carcinogenesis is highly heterogeneous [227,228,240]. Several miRNAs identified from such studies have been validated as tumor suppressors, although most of them appear unrelated to direct TP53 or RB pathways. In fact, typical genetic alterations (i.e., mutations, deletions, amplifications, and epigenetic modifications) are a common occurrence in cervical tumors at both miRNA loci or miRNA biogenesis effectors and regulators (i.e., DICER and DROSHA) [243,244]. Several studies and reviews indicate a myriad of miRNAs up- and downregulated in CC compared to normal cervical keratinocytes. It is, therefore, concluded that several variables play a role in determining miRNA disruption in cervical carcinogenesis.

To illustrate this last point, we present here a bioinformatic analysis using the Database of Differentially Expressed miRNAs in Human Cancers (dbDEMC) 3.0 (https://www.biosino.org/dbDEMC/index, accessed on 2 December 2024) and CancerMIRNome (http://bioinfo.jialab-ucr.org/CancerMIRNome//, accessed on 2 December 2024) to identify differentially expressed miRNAs (DEMs) between CC and normal uterine cervix samples [245,246]. The dbDEMC is an integrated database designed to store and display cancer-related miRNA data. The CancerMIRNome portrays the human miRNome of several cancer types from The Cancer Genome Atlas (TCGA). Both databases are extensively used to find gene expression differences in several cancer types. The microarray datasets EXP00166, EXP00167, and EXP00803, which include the comparison between normal and CC samples, were selected from the dbDEMC database. The CancerMIRNome database uses the “limma” package quantification software to perform the miRNA differential expression analysis [247]. As cutoff values, we selected an adjusted *p* < 0.05 and fold change > 1. A total of 194 DEMs were identified in CancerMIRNome. The intersection of the results showed 39 overlapping differentially expressed miRNAs between these two datasets. Additionally, 26 miRNAs were found to be upregulated and 13 downregulated (Figure 2). As in other studies, the TP53-responders miR-15b, miR-16, miR-200a/b/c, and miR-429, but not miR-34a/b/c, were found to be upregulated in CC. The miR-200 family has been validated for the promotion of cervical tumor cell proliferation through the regulation of the HIF-1α/VEGF signaling pathway and FOXG1 [248,249]. The origin of miRNA dysregulation found in this analysis is the different factors and altered proteins involved during all stages of CC. As an example, the expression of miR-130b is triggered by TNFα promoting cell cytotoxicity resistance [250]. Interestingly, hrHPV E6 and E7 upregulate miR-18a [251] and miR-16. E7 is responsible for the increased expression of miR-16-1 through the interaction of E2F with the promoter regions of *c-Myb* and *c-Myc*, activating the promoter region of the miR-15a/16-1 cluster [252]. Accordingly, the downregulation of miR-125b by HPV proteins promotes persistent infection and cervical cancer development [253], and HPV-16 E6 suppresses miRNA-23b expression in CC through DNA methylation of the miRNA host gene *C9orf3*, promoting CC development [254].

Through the functional enrichment analysis of the DEMs performed via miRTarBase on the CancerMIRNome, the targets of the DEMs were associated with the tumor-related biological process. The enrichment analysis of the upregulated miRNAs identified target genes involved in focal adhesion (GO:0048041) as *FN1*, *RHOA*, and *ROCK2*, in response to hypoxia (GO:0001666) as *ETS1* and *HIF1A*, in the cell cycle (GO:0071156) as *CDKN1A* and *MDM2*, involved in autophagy (GO:0010506) as *FOXK2*, and proteins involved in the production of miRNAs (GO:0035196) as LIN28B and DICER1. Secondly, the enrichment analysis for the downregulated miRNAs showed biological processes, such as apoptosis (GO:2001233), the cell cycle (GO:0044843), and mRNA catabolism (GO:0006402), including targets such as TP53, AKT1, BCL2, AGO2, and PUM1.

In a previous report, two other datasets were selected from the dbDEMC database (EXP00168 and EXP001669) to compare the miRNA expression profile of HSILs against CC. At least 28 miRNAs belonging to 7 miRbase V. 14 gene clusters at defined chromosomal loci showed concordant differential expression in both HSIL and CC [231]. Between these, miR-17-5p and miR-18a-5p (Cluster 5 at 13q31.3) and miR-20b-5p, miR-106a-5p, and miR-363-3p (Cluster 6 at 23q26.2-23q26.3) coincide with our own analysis as differentially expressed miRNAs in HSIL and CC. However, it has been challenging to find an miRNA signature in CC because the miRNAs deregulated vary among studies and in sample background, preventing the discovery of CC biomarkers.

## 7. Discussion

Many miRNAs can be found to be prevalently disrupted in cervical carcinogenesis. Thus, it is likely that a number of variables (i.e., sample origin, collection, and processing) may intervene in the outcome of the ultrasensitive NGS-related methodologies and platforms or bias the bioinformatic analysis algorithms. Also, it can be considered that the disruption of the miRNome in CC is a progressive phenomenon produced by the addition of individual and random changes due to the genomic instability of E6/E7-containing cells rather than a defined step within the cervical carcinogenesis pathway (Figure 3).

Noticeably, the amount of miRNAs with disrupted expression increases through cervical carcinogenesis pathways [227,228], thus indicating that a CC outcome may derive from accumulated changes in miRNA expression produced through sustained E6/E7 expression and their aforementioned effects on cell homeostasis. Whether the joint action of genomic instability and epigenetic changes leads to the malignization of E2-negative hrHPV-infected cells is still to be demonstrated. However, it has been shown that induced DNA methylation of hrHPV-containing immortal keratinocytes induces silencing of miRNAs, affecting anchorage-independent growth [255]. Furthermore, miRNAs found to be disrupted in progressive stages of cervical lesions are associated with several key hallmarks of cancer, such as growth suppressor evasion, replicative immortality, invasion and metastasis, cell death resistance, and sustained proliferative signaling [256].

Therefore, cervical carcinogenesis possesses a different dynamics than those of multi-step carcinogenesis in other tumor types (i.e., colon cancer). Of course, the initial necessary event for malignant CC is recurrent infection with hrHPV types. Nevertheless, this step by itself is insufficient for a carcinogenesis process. As reviewed above, the transient expression of hrHPV E6 and E7 oncogenes leads to the setup of hyperproliferation (aberrant and extended proliferation [103]) of the host cell and the inhibition of senescence (i.e., mTOR [45]) towards the replication of the viral genome (through E1 and E2 [65]), triggering an initial disruption of the miRNome, mainly through TP53-dependent miRNA expression (i.e., miR-34 [257,258] and miR-200 families [259]) and RB-dependent regulatory pathways (i.e., miR-92a [260], miR-93 [261]. This initial “hit” is usually transitory as an hrHPV productive infection requires terminally differentiated keratinocytes to produce and release infectious virions, and, thus, cell proliferation must cease [44]. Cancer risk assessment (CRA) at this point is uncertain as most infections (ASCUS, warts, or condylomata) will clear spontaneously [262], although the antiviral activity of APOBEC3 may result in mutagenic action on the cellular genome [214]. In a few cases, the abundance of viral genomes combined with a highly replicative environment results in the viral integration to the host genome, disrupting the E2 gene and resulting in the constitutive expression of E6 and E7. This event is a second hit that perpetuates and boosts E6/E7 functions, leading to further pro-carcinogenesis effects, now including Polycomb gene de-repression, epigenetic reprogramming, and p14^ARF^ and p16^INK4^ activation [211], causing unlimited proliferation but no malignization and clinical correlation with precursor lesions such as LSIL and HSIL. This, in turn, causes further disruption of the host miRNome, activating miRNAs related to sustained proliferation (i.e., miR-424 [263]), growth suppressor evasion (i.e., miR-199a [264]), and cell death resistance (i.e., miR-1 [265]). Here, the CRA is critical and should consider not only hrHPV typing but also p16^INK4^ and specific miRNome changes, plus an evaluation of the immune competence of the patient. It should be noted that at this stage, the process is still reversible [45,65,262].

As the clinical evolution continues from an HSIL to CC, the host cell undergoes an extensive amount of genetic and epigenetic changes as a consequence of the accumulation of genomic alterations and the activation of methylases and demethylases secondary to E6/E7 constitutive expression. It is unclear which HSILs will progress into CCs and which ones will not [2], but the increased miRNome expression might suggest a progression pattern or profile for carcinogenesis [266]. Moreover, epigenetic changes on particular miRNAs (miR-37, miR-124, miR-149, miR-203, and miR-375) have been suggested as specific markers for cervical cancer progression [255]. It should be noted that the infected keratinocyte retains a dependency on E6/E7 expression throughout the carcinogenesis process, as restoration of the TP53 and RB levels would inhibit the process in early stages, producing regression, or lead to senescence or apoptosis in HSILs and CCs [35,73]. However, although the accumulation of particular miRNAs has been consistently reported, an miRNA “cancer signature” for cervical carcinogenesis is yet to be described. The enormous variety of quantitative and qualitative miRNome changes appears to suggest a certain degree of individuality for each cervical tumor. That is, once an hrHPV-containing cell reaches an unlimited proliferation status, its fate is determined by the added action of random genetic and epigenetic changes over genes, transcripts, and proteins directly involved in sustained proliferation, cell death resistance, and growth suppressor evasion pathways, as well as extracellular factors, such as the tumor microenvironment and anti-tumor immunological responses, including immune suppression and inflammation. Therefore, CC would be the end result of hrHPV infection in a minority of patients as many other factors will play a critical role in cervical carcinogenesis. The TCGA project has identified recurrent mutations that ostensibly lead to invasive CC in *PIK3CA*, *FBXW7*, *MAPK1*, *PTEN*, *EP300*, *NFE2L2*, *CASP8*, *TK11*, *HLA-A*, and *HLA-B* genes, which are frequent in squamous CC, while *ELF3*, *CBFB*, *KRAS*, and *ARID1A* are enriched in cervical adenocarcinoma [267].

It is important to recognize that the additional changes necessary for HPV-induced cervical carcinogenesis may not be confined to genetic and epigenetic alterations within the cell but could also involve interactions with external factors. For instance, the microbiome, or bacterial composition, at specific body sites can significantly influence cancer development. Notably, during the progression from HPV-positive lesions to cervical cancer, there are changes in the cervicovaginal microbiome [268]. For example, 16s rRNA vaginal bacterial profiles of LSIL and CC patients present characteristics typical of bacterial vaginosis. These include a reduction in *Lactobacillus* populations, increased microbial diversity, and a higher prevalence of unusual anaerobic bacteria like *Sneathia sanguinegens* [269] and *Gardnerella vaginalis* [270]. The unique characteristics of bacterial vaginosis contribute to the weakening of immune and mucosal defenses against invading pathogens and their clearance. This occurs through various mechanisms, such as the release of enzymes that degrade mucin, disrupted pH balance, cytokine modulation, and persistent inflammation. These relationships warrant further exploration, including mechanistic studies, and could also be expanded to include analyses of the cervical virome or its viral composition [269]. How changes in the cervicovaginal microbiome impact the miRNome is not known, but recent advances in organ-on-a-chip models of the human cervix may lead to determining changes in the miRNome in response to hormonal and microbiome changes in a controlled environment [271].

**Figure 3 ijms-25-12991-f003:**
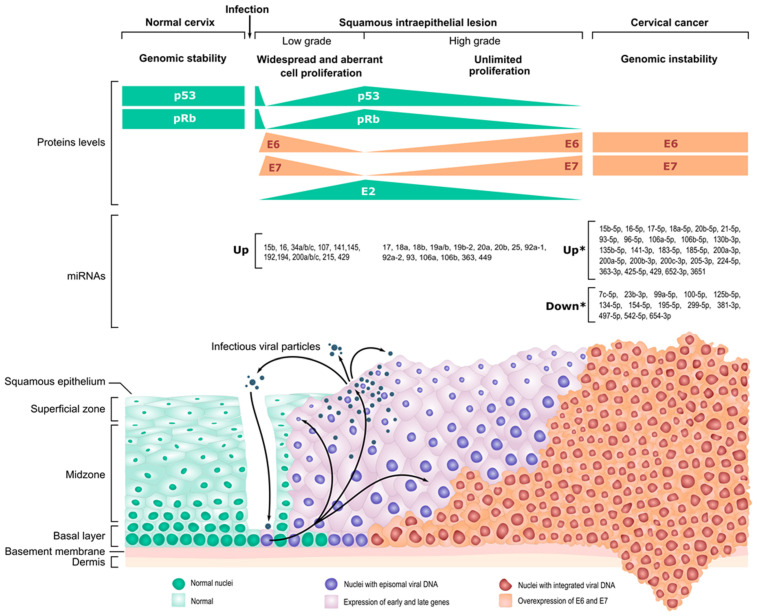
Natural history of HPV infection and cervical cancer overview. Significance: * *p* < 0.05. HPVs gain access to the epithelial basal layer through a microabrasion or infection of the transformation zone (green). The infected basal keratinocytes carry the viral genome in episomal form within the nuclei, where it transcribes and replicates. Initially, the early HPV genes are transiently expressed (see text). E6 and E7 proteins interact and degrade TP53 and RB, leading to widespread and aberrant cell proliferation to provide a suitable environment for viral maintenance replication and the dissemination of viral genomes through the actively mitotic basal layer. During this stage, and because of the activity of E6 and E7 proteins, several TP53-dependent miRNAs are inhibited, and E2F-regulated miRNAs are expressed, thus contributing to the hyperproliferative condition of the infected cell (purple). E2 expression eventually leads to repression of E6 and E7 transcription, thus allowing HPV genome replication in upper epithelial layers, where the replication rate increases to a vegetative state, and late genes L1 and L2 are expressed, leading to a productive infection and genital wart formation. In some instances, the co-existence of a highly replicative cellular environment and multiple HPV genome copies leads to viral DNA integration into the host cell genome. When the integration disrupts the E2 gene (red nuclei), the loss of E2 releases E6 and E7 expression, generating an unlimited proliferation state (orange) with the appearance of progressive degrees of genital dysplasia (LSIL to HSIL), which allows the accumulation of mutations due to the loss of TP53 and the manifestation of several other E6 and E7 functions, including epigenetic changes and the overall disruption of cell homeostasis. At this stage, the infected keratinocyte may gain an immune evasion phenotype, and the disruption of the miRNome widens quantitatively and qualitatively, leading to a tumor molecular landscape characterized by loss of control in cell regulation processes, which eventually leads to metastasis. Modified from [272].

## 8. Conclusions

As the hrHPV infection sets up the conditions for a highly proliferative environment conducive to sustaining viral replication for long periods of time, the host cell genome is drawn to a transient mutation-prone condition due to the functional absence of TP53, APOBEC3 activation, and a highly replicative intracellular environment. A persistent hrHPV infection activates a number of miRNAs under TP53 and E2F regulation as a result of the combined E6/E7 activity. The infected keratinocyte responds through activation of APOBEC3, which triggers an immune response against the virus but also introduces mutations into the viral genome and, collaterally, on the host genome. If untreated, the infection may result in the integration of hrHPV DNA into the host genome, disrupting the E2 gene, thus perpetuating proliferation, allowing for a permanent and constitutively unstable molecular landscape, and leading to a pro-carcinogenic condition that is enhanced by activated oncogenes and regulatory ncRNAs, such as miRNAs. Therefore, the increasing presence of miRNAs during cervical carcinogenesis is a direct result of the combined activities of viral and cellular factors, which, in turn, rely on the tumor microenvironment, epigenetics, and immune surveillance status in every infected individual. Although no particular miRNome profile has been linked to cervical carcinogenesis risk, it is clear that the increase in overall miRNA expression is indeed a measure of the malignant status of the cell and could be considered a risk factor but not a cause. Aside from HPV typing and clinical classification, CRA for hrHPV infection must consider the particular miRNome and immunological status of the patient.

It should be noted that throughout the whole cervical carcinogenesis process, there is a reliance on E6 and E7 expression, regardless of the miRNome and immunological profile of the infected cell. This feature remains the main rationale behind therapeutic vaccination and anti-E6/E7 technologies as potential CC treatments. However, prophylactic vaccination against hrHPV infection and anti-PD-1 therapy may be considered the most important steps towards invasive CC prevention and treatment. Future studies on miRNomics (including miRNA epigenomics and circulating miRNAs), APOBEC3 expression, and immune status will lead to the establishment of high-risk patient profiles in addition to hrHPV typing as the most valuable tools for CC early detection and prevention.

## Figures and Tables

**Figure 2 ijms-25-12991-f002:**
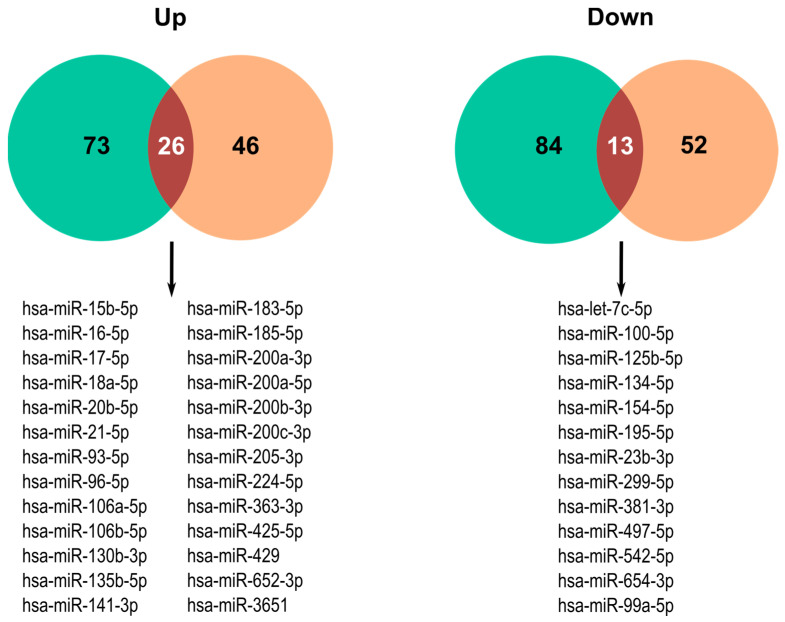
Venn diagram of differentially expressed miRNAs in cervical cancer. CancerMIRnome and dbDEMC databases were used to determine the differentially expressed miRNAs between normal cervical epidermis and CC samples. Cutoff values for the analysis and adjusted *p* were < 0.05 and fold change > 1, respectively. The names of the significantly deregulated miRNAs in both databases are listed.

**Table 1 ijms-25-12991-t001:** MiRNAs reported to be expressed in cervical carcinogenesis. SIL, squamous intraepithelial lesion; CC, cervical carcinoma.

Stage	Upregulated miRNA	Downregulated miRNA	Refs.
SIL	miR-215, miR-449 miR-17, miR-18a, miR-19a/b, miR-20a, miR-92a-1, miR-18b, miR-19b-2, miR-20b, miR-92a-2, miR-106a, and miR-363 and miR-25, miR-93, miR-106b	miR-15b, miR-16, miR-34a/b/c, miR-107, miR-141, miR-145, miR-192, miR-194, miR-200a/b/c	[227,228,238,239]
CC	let-5p, miR-9-5p, miR-10a-5p, miR-15b, miR-16-5p, miR-20, miR-21-5p, miR-25-5p, miR-27a, miR-29, miR-30, miR-31, miR-92b, miR-92a-3p, miR-93, miR-96, miR-106a, miR-125b, miR-127, miR-133a/b, miR-135b, miR-141a/b, miR-142, miR-146, miR-150, miR-155, miR-181, miR-182, miR-185, miR-189, miR-193a-3p, miR-196a, miR-199a/b, miR-203a/b, miR-205, miR-210, miR-215, miR-221, miR-222, miR-223, miR-224, miR-301b, miR-320, miR-361, miR-373, mi-R378, miR-425, miR-449, miR-451a, miR-466, miR-486-5p miR-494, miR-500, miR-505, miR-519d, miR-543, miR-590-5p, miR-711, miR-720, miR-886-5p, miR-888, miR-892b, miR-944, miR-1246, miR-1285, miR-1290, miR-2392, miR-3147, miR-3162, miR-4484, miR-6852	let-7a/b/c/g, miR-1, miR-7; miR-10b, miR-17-5p, miR-22, miR-23b, miR-24, miR-26a, miR-27b, miR-29a/b, miR-30a/e, miR-99a/b, miR-100-5p, miR-101, miR-103b, miR-107, miR-124-3p, miR-125a/b, miR-126-3p, miR-129b-5p, miR-132, miR-133a, miR-138, miR-139-3p, miR-140-5p, miR-141, miR-142-3p, miR-143, miR-144, miR-145, miR-149, miR-152, miR-154, miR-181, miR-182, miR-183, miR-186, miR-187, miR-193a/b, miR-195, miR-196b-5p, miR-199a/b, miR-202, miR-203, miR-204, miR-205, miR-211, miR-212, miR-214, miR-216-5p, miR-218, miR-223, miR-296, miR-320; miR-326, miR-328, miR-329, miR-331-3p, miR-335, miR-337, miR-338-3p, miR-342, miR-362, miR-374c-5p, miR-375, miR-376a/c, miR-379, miR-383, miR-424, miR-429, miR-451, miR-484, miR-486-3p miR-489-3p, miR-491-5p, miR-494, miR-497, miR-503, miR-506, miR-544, miR-630, miR-634, miR-638, miR-720, miR-758, miR-892b, miR-1297, miR-1246, miR-2861, miR-3156-3p, miR-3185, miR-3666, miR-3960, miR-4262, miR-4467, miR-4488, miR-4525	[227,228,238,240,241,242]

## Data Availability

The sources of all data used in this publication is open access and freely available and has been cited throughout the manuscript.

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
