# Peer review of "The Natural History of Cervical Cancer and the Case for MicroRNAs: Is Human Papillomavirus Infection the Whole Story?"

_ijms, 2024, doi:10.3390/ijms252312991_

Round 1
Reviewer 1 Report (Previous Reviewer 2)
Comments and Suggestions for Authors
The article submitted for review is now much more clearly formulated and the figures more clearly reflect the focus, which consists of a very up-to-date and detailed summary of the HPV genome and its role in viral replication and transmission to neoplasia and even cancer. In the second main part, as I read the manuscript, the authors summarize the influence of miRNAs on cell processes with a focus on tumorigenesis in general and hrHPV viruses in particular. I consider the paper to be a valuable overview of the topic “Processes of tumorigenesis in HPV” and “Comprehensive overview of the regulation of miRNAs in cervical cancer”. Although, as the authors themselves write, the actual functional miRNAs of HR-HPV have yet to be identified. The authors fail to convey their message, which I understand to be that they see miRNomics (including miRNA epigenomics and circulating miRNAs) as an essential key to clinically highly relevant high-risk patient profiling, alongside genomics, transcriptomics, APOBEC3 expression and the patient's immune status. However, I think the review is really worth publishing. However, the argumentative thread should be worked out more clearly in order to understand the part in the title “Is HPV infection the whole story?”.
Perhaps a reorganization of the sections would help here. My suggestion, which I would like to put to the authors for discussion, is: Section 3 “MicroRNAs (miRNAs)” (line 95-ff) should be positioned after Section 5 “HPV den cevial cancer (line 556-714 in my pdf file). It can be subsumed under section 6: “MIRNAs in Cervial Cancer”. The importance of the miRNome in clinical risk assessment could also be pointed out here.
Otherwise, there are only minor corrections that do not change the actual content:
Lines 33-34: „…is critically reviewed in the context of hrHPV infection natural history cervical cancer.” There is something wrong with the sentence.?
Line 48: The abbreviation HIV should be written out in full
Line 81: The reference to an illustration is unusual in the introduction, especially as it is no. 3 of the illustrations.
Line 318: It should be: “amino acids”
Author Response
We thank the reviewer for the comments
Lines 33-34: „…is critically reviewed in the context of hrHPV infection natural history cervical cancer.” There is something wrong with the sentence.?
We corrected the sentence. "is critically reviewed in the context of the natural history of hrHPV infection and cervical cancer."
Line 48: The abbreviation HIV should be written out in full
We write the full name
Line 81: The reference to an illustration is unusual in the introduction, especially as it is no. 3 of the illustrations.
We removed the reference to figure 3 from the introduction
Line 318: It should be: “amino acids”
We corrected the sentence
Reviewer 2 Report (Previous Reviewer 3)
Comments and Suggestions for Authors
Dear Authors,
This article titled " Cervical Cancer: Is HPV the Whole Story? " The article is well-structured, covering the HPV life cycle, E6/E7 oncoproteins, and the miRNA profile in cervical cancer. In additionally, authors addressd all of my questions in pervious version. I have no more question.
Author Response
We thank the reviewer
Round 2
Reviewer 1 Report (Previous Reviewer 2)
Comments and Suggestions for Authors
All minor critic points were fulfilled.
The authors did not reposition Section 3 “microRNAs”, but I left that up to them. However, I maintain that this point seems somewhat lost between Section 2 “Cervical cancer” and the very detailed Section 4 “Human papillomaviruses”.
The question raised in the title “Is HPV infection the whole story?! still remains unanswered for me.
This manuscript is a resubmission of an earlier submission. The following is a list of the peer review reports and author responses from that submission.
Round 1
Reviewer 1 Report
Comments and Suggestions for Authors
The role of miRNAs in HPV-induced carcinogenesis and cervical cancer is very relevant in the field. Nevertheless, although reviews on this topic are needed, the study by Palomino-Vizcaino falls short in emphasizing on the miRNA-HPV-cervical cancer relationship and in-depth revisions are required.
1. The focus of the review is not clear in the abstract, introduction, or main text. The issue of the journal and keywords of the paper suggest the focus is on miRNAs and ncRNAs, but the study only emphasize this in Line 24 in the abstract and starts this discussion in Line 504 in the main text. The information between Lines 31 and 504 should be heavily edited and summarized, to emphasize the focus on miRNAs.
2. The details between Lines 618 and 640 need a table for better visualization. This section is not appealing and/or easy-to-understand for readers.
3. The authors perform bioinformatic analysis in this review (Line 643), which further highlight the lack of focus in the paper and the manuscript type submitted.
4. Several references are older than 25 years ago, outdated, and imply that the review is not sufficiently relevant to be published. Several relevant recent papers (2019 onwards) on miRNAs (using cervical scrapes, self-swabs, urine) and HPV biology (HPV genotypes and integration) are missing.
5. Please use the right scientific terminology in the HPV field, such as LSIL, HSIL, hrHPV, lrHPV.
6. About 50% of ASCUS cases progress to LSIL, so the carcinogenesis timeline implied in Lines 44-48 is incorrect. Please update.
7. Please avoid referring to ISCC, and use HSIL and CC. This is outdated. Line 57.
Reviewer 2 Report
Comments and Suggestions for Authors
The article submitted for review deals with HPV-induced cervical cancer. The article begins with a classification of the current incidence of cervical cancer, presents the viral aetiology and clinical stages of the disease and lists possible treatment options.
A significant part of the work contains a very comprehensive and well-structured overview of the genetic structure of HPV viruses, as well as the functions of the individual genes in viral replication and the development of neoplasia and cancer. The focus here is on the genome of the HR-HPV viruses, although it is not clear whether the individual aspects apply only to the HR-HPV viruses and distinguish them from the LR-HPV viruses. The complexity of the degeneration of the keratinocytes after infection with HR-HPV and the actual development of tumors is explained in detail. As modulating factors in tumorigenesis, non-coding cRNAs (here especially the micro RNAs) are comprehensively listed in the article and their known role in tumorigenesis in general and in HPV-induced tumorigenesis in particular is systematically summarized. The stages of tumorigenesis from infection with HPV viruses (HR-HPV), the key players involved in the individual stages of squamous intraepithelial lesions through to malignant cervical cancer are illustrated very clearly in Fig. 3.
One major point of criticism that arose when reading through the manuscript is that the question raised in the title “Is HPV the whole story?” is not answered. Here I see 2 options. 1) The title should be changed to reflect the central points of the review 2) The question should at least be answered in the summary so that the reader knows what the authors mean here.
Further remarks are:
Throughout the text: All gene names and symbols should be written in Italic (e.g., BRCA1, TP53). The protein product should not be italicized (e.g., BRCA1 protein
Line 8:
Line 63:
Reference (3, Arbyn, et al, 2008). Isn ́t there any newer reference such as: Li Y et al., 2017
Li Y, Xu C. Human Papillomavirus-Related Cancers. Adv Exp Med Biol. 2017;1018:23-34. PMID: 29052130.
Reference (8, Alvarez-Salas et al,. 2008). This self-quotation is also relatively old and does not take into account the more recent therapeutic possibilities. The assertion that there are no better alternatives in treatment is simply not correct. Current literature should be listed here, such as Immunotherapies using Check-point inhibitors (for example: Yang et al,. 2024; Farajimakin, 2024; Grau el al 2023) or other treatments such as photodynamic therapy, oral medications, and non-surgical device treatments McGee et al,. 2024)
Yang ST, Wang PH, Liu HH, Chang CW, Chang WH, Lee WL. Cervical cancer: Part II the landscape of treatment for persistent, recurrent and metastatic diseases (I). Taiwan J Obstet Gynecol. 2024 Sep;63(5):637-650. PMID: 39266144.
Farajimakin O. The Role of Immunotherapy in the Treatment of Gynecologic Cancers: A Systematic Review. Cureus. 2024 Jul 29;16(7):e65638. PMID: 39205726.
JIMS-324525
Grau JF, Farinas-Madrid L, Garcia-Duran C, Garcia-Illescas D, Oaknin A. Advances in immunotherapy in cervical cancer. Int J Gynecol Cancer. 2023 Mar 6;33(3):403- 413. PMID: 36878562.
McGee AE, Hawco S, Bhattacharya S, Hanley SJB, Cruickshank ME. Alternatives to surveillance for persistent human papillomavirus after a positive cervical screen: A systematic review and meta-analysis. Eur J Obstet Gynecol Reprod Biol. 2024 Nov;302:332-338. PMID: 39369502.
Line 129:
Line 138:
Line 449: I think, there is a mistake: E43
Line 492: The Reference Lyng et al., 2014 is missing in the reference list
Lines 493-496: This assertion should be supported by a literature reference.
Line 504: There is “as” missing. (“in several cellular processes such as apoptosis,...”
Line 618-ff: Is it really necessary or useful to list all regulated miRNAs in CC if a filtered overview (at least that's how I understand it) is shown in Fig. 2.? The text passage is unreadable and redundant.
Line 656: Please write out the abbreviation (DEM).
Fig.3 and figure legend: Only the expression of E6 and E7 is shown in the cartoon, but not E1, E2, E4 and E5 as correctly stated in the legend (line 763). Line 765: “...in synchrony with E1, E2 765 and E4 expression provide a suitable...” This fact is also not shown in the figure. Line 770-771: “...infected keratinocytes to move up 770 the midzone of ...“ This implies, that reaching the superficial zone is an active process by the infected cell. Is this the case?

Reviewer 3 Report
Comments and Suggestions for Authors
This article titled " Cervical Cancer: Is HPV the Whole Story? " The article is well-structured, covering the HPV life cycle, E6/E7 oncoproteins, and the miRNA profile in cervical cancer. However, the sections sometimes dive deeply into technical aspects, which can make the central message less clear. Reorganizing some parts for clarity and focusing on how each section contributes to understanding non-HPV factors would benefit the readers. The manuscript is well written and interesting. However, this manuscript cannot be accepted in this version. My comments are described as followings:
Comments:
1. In the abstract section, the abstract effectively summarizes the role of HPV and introduces non-coding RNAs. However, it could be enhanced by briefly outlining other potential co-factors in cervical carcinogenesis to align with the title.
2. In the introduction section, this section provides an excellent background on HPV and cervical cancer but could briefly introduce non-HPV-related factors or unknown pathways involved in the disease. The statement that cervical cancer remains a major cause of death in developing countries is important; adding recent statistics would make it more impactful.
3. Authors detailed review and summarize the HPV mechanisms. Sections on HPV’s E6/E7 proteins and their role in tumorigenesis are detailed and comprehensive. However, they could benefit from streamlined discussion that better highlights their relevance to the article's broader question about HPV being the sole cause. On the other hand, figures are used effectively to illustrate the HPV life cycle. More figures detailing the interaction of HPV with other potential etiological factors could be considered.
4. The inclusion of miRNA as a topic is a strong point, showing how HPV infection influences miRNA expression in cervical cancer. This section would benefit from additional details on how these miRNAs might interact with non-HPV factors. However, some of the references to databases (like dbDEMC and CancerMIRNome) are insightful but could be clarified to explain their relevance to the study's objective more effectively.
5. HPV vaccine is critical on cervical cancer prevention; however, this issue is missing in this article.
6. The discussion primarily recaps findings about HPV and cervical cancer without fully addressing potential non-HPV factors as the title suggests. Adding a few paragraphs on other potential cofactors—such as genetic predispositions, immune response variability, and socio-environmental influences—would broaden the scope and improve alignment with the title.
7. The conclusion is concise but could be expanded to include implications for future research directions, specifically targeting gaps in understanding non-HPV contributions to cervical cancer.
8. Additional tables or figures summarizing the pathways influenced by HPV versus those potentially independent of HPV would enhance comprehension and engagement.
9. Some sections, such as those detailing specific proteins and miRNAs, could be streamlined to maintain a more accessible read while retaining critical details.
10. Consider a title revision to reflect the heavy focus on HPV if non-viral factors are not further discussed. Alternatively, more content could be added to the manuscript on non-HPV etiological contributors.
Comments on the Quality of English Language
Minor English Editing is requored.